# Dream Recall and Content versus the Menstrual Cycle: A Cross-Sectional Study in Healthy Women

**DOI:** 10.3390/medsci7070081

**Published:** 2019-07-21

**Authors:** Ioannis Ilias, Nicholas-Tiberio Economou, Anastasia Lekkou, Andrea Romigi, Eftychia Koukkou

**Affiliations:** 1Department of Endocrinology, Elena Venizelou Hosp, 11521 Athens, Greece; 2Sleep Study Unit, Eginition Hospital, University of Athens & Enypnion Sleep Disorders—Epilepsy Center, 11521 Athens, Greece; 3IRCCS Neuromed, Sleep Medicine Center, 86077 Pozzilli, Italy

**Keywords:** sleep, dreams, women, menstrual cycle

## Abstract

The association between sleep and the menstrual cycle has been scarcely studied. This study aimed to investigate the association between dream recall and content and the menstrual cycle among a large sample of young women. To this aim, 944 women were asked about their day of menstrual cycle, whether they remembered the previous night’s dreams and if they did so to describe the dream content as pleasant or unpleasant. A total of 378 women recalled the previous nights’ dreams, with 199 reporting pleasant dream affect/content and 179 reporting unpleasant dream content. In women who recalled their dreams, there was an association of pleasant dream content with the luteal phase (*p* = 0.038). In conclusion, in women, the hormonal milieu of the luteal phase may influence dream content.

## 1. Introduction

Sleep may be affected by variations in reproductive hormones, stress, depression, aging, life/role transitions, and other factors. The menstrual cycle is associated with changes in the sleep-wake cycle. Menstruating women with or without significant menstrual-related complaints often report poor sleep quality and sleep complaints during the premenstrual week compared to other times of the menstrual cycle [1]. Dream interpretation—necessitating dream recall—is a human universal [2]. Dream recall and content apparently follow individual/personal traits (such as memory and personality) [3]. Few robustly evaluated correlates of dream recall and content have been delineated [3]. Conflicting results have been presented regarding dream recall and dream affect vis-à-vis the menstrual cycle [4,5,6]. Progestagens in contraceptive pills may improve dream recall [7]. The relevant studies available have had few subjects studied over weeks or months. With this study we aimed to assess dream recall/affect vis-à-vis the menstrual cycle in a large sample of women.

## 2. Materials and Methods

The women were consecutively recruited—on a volunteer basis—from health sector employees and health sciences students at the Elena Venizelou Hospital (Athens, Greece). They reported no indication of sleep or medical disorders. All the volunteers were of normal psychological status, as assessed by psychiatric interview. The women did not complain of dysmenorrhea. We studied only women with self-reported regular menstrual cycles of 21–35 days in a cross-sectional fashion. We considered that the luteal phase of the menstrual cycle was relatively stable for all women, with a duration of 14 days [8]. We excluded women with irregular menstrual cycles or with shorter/longer cycles, symptoms of premenstrual syndrome and premenstrual dysphoric disorder or women under systemic pharmacotherapy of any kind. The subjects were given an ad hoc simple anonymous questionnaire with the following items: Age, menstrual cycle duration, day of their menstrual cycle, dream recall regarding the previous night’s dreams and dream affect/emotional content (only described as positive/pleasant or negative/unpleasant). The study was approved by the Local Ethics Committee of the Hospital (No 01/2017, 25 January 2017). Each subject signed a written informed consent regarding the objective and the goal of the study. Statistical analysis was performed by logistic regression, with dream recall and content as dependent variables, age as a continuous variable and menstrual phase (follicular or luteal) as a factor; *p* values < 0.05 were considered statistically significant (MedCalc v14, MedCalc, Oostende, Belgium).

## 3. Results

We studied 944 healthy women (mean age ± SD: 31.3 ± 9.0 years; range 17–51 years). According to the subjects’ responses 453 women were in the follicular phase and 491 in the luteal phase. A total of 378 women recalled the previous nights’ dreams, with 199 reporting positive/pleasant dream affect/content (94 in the follicular and 105 in the luteal phase) and 179 reporting negative/unpleasant dream affect/content (105 in the follicular and 74 in the luteal phase). Age was not associated with dream recall or content and dream recall was not associated with the menstrual phase. However, in women who recalled their dreams, there was an association between dream content and menstrual phase (and more in detail, of those with a positive affect with the luteal phase) (Table 1).

## 4. Discussion

We tried to assess the relationship of the menstrual cycle vis-à-vis dream recall and content with a questionnaire-based study design in a large sample of healthy women and without having to deal with the issue of the subjects’ motivation or training in reporting their dreams. Research has shown that keeping a dream diary increases dream recall [9]. We opted for a “barebones” basic retrospective questionnaire to facilitate the subjects’ answers, avoiding the use of any long retrospective questionnaire with its associated underestimation of the dream recall problem [10]. Thus, we avoided the use of cumbersome dream journals or of assessing reports of many days, weeks or months [11,12]. Some people remember their dreams more easily than others [13,14,15]. Moreover, dream recall has been reported to decline with age [16,17]; no such effect of age was observed in our subjects, possibly due to the limited age range of the women who were included. Although dream recall per se was not found to be associated with the menstrual cycle, in this large sample of women that recalled the previous nights’ dreams a tendency towards pleasant dream content was noted in the luteal phase. This is apparently linked to the hormonal milieu of the luteal phase, which is characterised by the increased secretion of progesterone (the latter may be implicated in the consolidation of memories [18] and may also alter dreaming during pregnancy [19,20]) but also of altered dynamics in other hormones (see below). Dream recall is reported to be highest for REM dreams [21], thus our finding might be of further importance, given the possible changes in REM sleep frequency, noted as a result of current lifestyle conditions [22]. Cortisol may also have an impact on how emotions are experienced in the process of dream formation [23]; that is, during sleep, cortisol levels peak in the REM phases [23]. It has been hypothesized that cortisol may be implicated in dream recall (by fragmenting the phenomenological dream experience and disrupting memory retrieval) [23]. Interestingly, the cortisol response in laboratory-induced mental stress may be enhanced in the luteal phase [24]. Thus, based on our findings, we can also suggest that cortisol in the luteal phase may assist in dream amnesia, contributing to this “sieve” function and of selectively constraining unpleasant dream memories (and letting the “passage” of pleasant dreams).

We have to acknowledge the limitations of our study. A bidirectional mutual association links daytime emotions and dreams [11,25]. Acute or chronic emotional stress may have a differentiated effect on sleep by influencing sleep physiology and dream patterns, dream content and the emotion within a dream [26]; in this study the effect of stress was not assessed. Another notable omission is that of several sociodemographic variables that have also been associated with differences in sleep patterns (which in turn may influence dream content), such as lower educational attainment, depression, unemployment, physical activity, marital status and perception of social support [27]. Regarding the education level of our subjects, all were either health sciences students or had at least a college degree and working in a healthcare setting (nevertheless these sample subjects’ characteristics were not systematically noted).

It is evident that sleep development patterns along women’s lifetime are complex. A thorough evaluation of associations among sleep, dream content and biological/physiological processes should take into account physical, mental health and life circumstances. Further analyses of dream content along with assessment of sleep architecture and fluctuations in hormones such as progesterone or cortisol levels could be performed for shedding light on the physiological changes and dream themes/content noted in women in different menstrual phases.

## Figures and Tables

**Table 1 medsci-07-00081-t001:** Logistic regression results regarding positive dream content/affect (*n* = 378) ^1^.

Variable	B	SE	*P*	OR	95% CI
Age	−0.007	0.012	0.526	0.992	0.969–1.016
Luteal phase (+)	+0.451	0.217	0.038	1.569	1.025–2.402

^1^ Hosmer–Lemeshow chi square: 4.544, *p* = 0.585; B: b coefficient; SE: standard error; *P*: probability; OR: odds ratio; 95% CI: 95% confidence interval.

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
