# Peer review of "Dream Recall and Content versus the Menstrual Cycle: A Cross-Sectional Study in Healthy Women"

_medsci, 2019, doi:10.3390/medsci7070081_

Reviewer 1 Report

This report tried to assess the relationship of the menstrual cycle and dream recall and content among a sample of 944 healthy women. Among women who remembered their dreams, an association was found between “pleasant dream” and the luteal phase. The topic is very interesting and promising, but even though the finding may be significant, the study suffers from poor methodology. Many potentially confounding variables are left out. For instance, it is well known that emotional stress may have a differentiated effect on sleep by influencing sleep physiology and dream patterns, dream content and the emotion within a dream. Furthermore, several sociodemographic variables have also been associated with differences in sleep patterns which in turn may influence dream content, such as lower educational attainment, depression, unemployment, physical activity, marital status and perceived social support. Given the complexity of sleep development patterns along women’s lifetime, clinical practices should assess physical, mental health and life circumstances when evaluating an association between sleep, dream content and biological processes.

 Author Response

1.a. This report tried to assess the relationship of the menstrual cycle and dream recall and content among a sample of 944 healthy women. Among women who remembered their dreams, an association was found between “pleasant dream” and the luteal phase. The topic is very interesting and promising, but even though the findings may be significant, the study suffers from poor methodology.

The reviewer is right to point the limitations of the study regarding methodology. 
It is true that we opted for a "barebones" basic retrospective questionnaire to facilitate the subjects' answers (and avoiding the use of any long retrospective questionnaire with its associated underestimation of dream recall problem. We acknowledge in the Discussion the limitations of this study, mainly the non-inclusion of parameters with an effect on dream recall and content.

We added the following to the Discussion 
We opted for a "barebones" basic retrospective questionnaire to facilitate the subjects' answers (and avoiding the use of any long retrospective questionnaire with its associated underestimation of dream recall problem. 

Furthermore, we acknowledge in the Discussion the limitations of this study, mainly the non-inclusion of parameters with an effect on dream recall and content (see below)

1.b. Many potentially confounding variables are left out. 

The reviewer is right that many confounding variables were left out of this study. We acknowledge this in the limitations of the study (see also point 1.a. and our response to it above).

1.c. For instance, it is well known that emotional stress may have a differentiated effect on sleep by influencing sleep physiology and dream patterns, dream content and the emotion within a dream. 

Following on point 1.b. we present these limitations in the Discusssion, adding the relevant references wherever appropriate and possible. The following text was added:
A bidirectional mutual association links daytime emotions and dreams. Acute or chronic emotional stress may have a differentiated effect on sleep by influencing sleep physiology and dream patterns, dream content and the emotion within a dream; in this study the effect of stress was not assessed.

1.d. Furthermore, several sociodemographic variables have also been associated with differences in sleep patterns which in turn may influence dream content, such as lower educational attainment, depression, unemployment, physical activity, marital status and perceived social support. 

Also following on point 1.b. we present these limitations in the Discusssion, adding the following text:
Another notable omission is that of several sociodemographic variables that have also been associated with differences in sleep patterns (which in turn may influence dream content), such as lower educational attainment, depression, unemployment, physical activity, marital status and perception of social support. Regarding the education level of our subjects all were either health sciences students or had at least a college degree and working in a healthcare setting (nevertheless these sample subjects’ characteristics were not systematically noted).

1.e. Given the complexity of sleep development patterns along women’s lifetime, clinical practices should assess physical, mental health and life circumstances when evaluating an association between sleep, dream content and biological processes.

The reviewer is correct in pointing out these factors as having an effect on dream recall and content. We added the following paragraph to the Discussion section: 
It is evident that sleep development patterns along women’s lifetime are complex. A thorough evaluation of associations among sleep, dream content and biological/physiological processes should take into account physical, mental health and life circumstances.

Reviewer 2 Report

The study aimed to assess dream recall/affect vis-à-vis the menstrual cycle in a large sample of women. Although dream recall per se was not found to be associated with the menstrual cycle, there was an association of dream content with menstrual phase. The topic of this brief report is good and present interesting results. We only have two minors concerns :

1) The introduction could be more elaborated as the conceptual background of the study

2) In the discussion, the authors should discuss briefly the limitation of their study.

Thank you for the opportunity to review this interesting brief manuscript.

Author Response

2.a. The study aimed to assess dream recall/affect vis-à-vis the menstrual cycle in a large sample of women. Although dream recall per se was not found to be associated with the menstrual cycle, there was an association of dream content with menstrual phase. The topic of this brief report is good and present interesting results.

We thank the reviewer for conveying his/her positive impression of this study.

2.b. The introduction could be more elaborated as the conceptual background of the study

We have added the following to the paper's Introduction: 
Dream interpretation - necessitating dream recall - is a human universal. Dream recall and content apparently follow individual/personal traits (such as of memory and personality). Few robustly evaluated correlates of dream recall and content have been delineated.

2.c. In the discussion, the authors should discuss briefly the limitations of their study.
Following this reviewer's suggestion along with the suggestions of reviewer 1 we have expanded the limitations of the study section, also adding some new references to substantiate these additions.

 Round  2

Reviewer 1 Report

the improved version is suitable for publication